



# Emission dynamics of reactive oxygen species and oxidative potential in particles from a gasoline car and wood stove

Battist Utinger[*,1], Alexandre Barth[*,1], Andreas Paul[2], Arya Mukherjee[3], Steven J. Campbell[4], Christa-Maria Müller[1], Mika Ihalainen[3], Pasi Yli-Pirilä[3], Miika Kortelainen[3], Zheng Fang[5], Patrick Martens[6], Markus Somero[3], Juho Louhisalmi[3], Thorsten Hohaus[2], Hendryk Czech[6,7], Olli Sippula[3,8], Yinon Rudich[5], Ralf Zimmermann[6,7], and Markus Kalberer[1]

[1]Department of Environmental Science, University of Basel, Basel, Switzerland
[2]IEK-8 Troposhere, Forschungszentrum Jülich GmbH, Jülich, Germany
[3]Department of Environmental and Biological Science, University of Eastern Finland, Kuopio, Finland
[4]MRC Centre for Environment and Health, Environmental Research Group, Imperial College London, London, UK
[5]Department of Earth and Planetary Science, Weizmann Institute of Science, Rehovot, Israel
[6]Department of Technical and Analytical Chemistry, University of Rostock, Rostock, Germany
[7]Cooperation group "Comprehensive Molecular Analytics", Helmholtz Centre Munich, Munich, Germany
[8]Department of Chemistry, University of Eastern Finland, Joensuu, Finland

*both authors equally contributed to the publication

Correspondence to: Markus Kalberer (markus.kalberer@unibas.ch)

**Abstract.** Air pollution is one of the largest environmental health risks and one of the leading causes of adverse health outcomes and mortality worldwide. The possible importance of the oxidative potential (OP) as a metric to quantify particle toxicity in air pollution is increasingly being recognized. In this work, the OP and reactive oxygen species (ROS) activity of particles from fresh and aged gasoline passenger car emissions and residential wood combustion (RWC) emissions were investigated using two novel instruments. Applying online instruments using an ascorbic acid (AA) and 2',7'-dichlorodihydrofluorescein (DCFH) assay provides a much higher time resolution compared to traditional filter-based methods and allows for new insights into highly dynamic changes in OP and ROS activity of these sources. Due to the efficiency of the particulate filter in the Euro 6d car, almost no primary particles were emitted and thus no particle OP and ROS was detected in primary exhaust. However, a substantial and highly dynamic OP and ROS activity was observed after photochemical ageing due to the formation of secondary particles. Increasing OP and ROS activity due to ageing was also observed when comparing fresh and aged RWC emissions. Overall, RWC emissions had significantly higher OP and ROS signals compared to car emissions. This suggests that aged RWC emissions could be a major contributor to air pollution toxicity, and may be an intrinsically more harmful emission source than car exhaust, although the formation potential for secondary particles from car emissions was still high. These measurements illustrate the strong differences and highly dynamic nature of toxicity-relevant particle properties from two air pollution sources and could contribute to more efficient air pollution mitigation policies.



## 1   Introduction

Air pollution is a complex mixture of gaseous compounds and aerosol particles composed of both natural and anthropogenic constituents, with common anthropogenic sources including fossil fuel combustion, non-exhaust traffic emissions, industrial processes, and agricultural activities. The specific composition and concentrations of pollutants in the ambient air varies depending on a range of factors like local sources and meteorological conditions. According to numerous epidemiological studies, there is a compelling association between air pollution and various health effects.(Hart et al., 2015; Laden et al., 2006; Lepeule et al., 2012) Elevated levels of ambient aerosol particles have been associated with increased hospital admissions and deaths from a variety of diseases, including cancer, respiratory illness, and cardiovascular disease.(Baulig et al., 2003; Donaldson et al., 2001; Li et al., 2003; Offer et al., 2022; Prahalad et al., 2001) The World Health Organization (WHO) recognizes in a recent report that air pollution and specifically aerosol particles are the single largest environmental threat to human health.(World Health Organization, 2021) Despite strong evidence linking air pollution particles and negative health outcomes, the specific mechanisms and properties by which aerosol particles cause these effects are poorly understood. Identifying the health-relevant particle properties, including the most damaging chemical components and their emission sources, will allow for the development of more effective strategies to minimize exposure from the most harmful sources.(Brunekreef and Holgate, 2002; Künzi et al., 2015) Current guidelines and legal regulations mainly address total particle mass concentration as the criterion to evaluate air quality. However, there is proof that differences in the chemical composition of particulate matter play a role in their toxicity.(Kelly and Fussell, 2012) There is increasing evidence that oxidising particle components might play a role in particle toxicity.(Øvrevik et al., 2015)

Reactive oxygen species (ROS) are highly reactive oxygen-containing molecules and radicals such as hydrogen peroxide, hydroxyl radical, superoxide, and organic peroxides and are formed in oxidation reactions in the atmosphere. These can be delivered exogenously through particle exposure (especially larger, less volatile organic peroxides and radicals) or can be produced endogenously when particles are deposited on the lung surface after inhalation such as hydroxyl radical or superoxide.(Campbell et al., 2019) Another potential metric serving as proxy for particle toxicity is the oxidative potential (OP) of particles, which is defined as the ability of aerosol particle components to generate ROS inside the body while simultaneously deplete antioxidants.(Kelly, 2003) The exceedance of the body's anti-oxidative capacity can lead to oxidative stress which has been linked to causing negative health effects.(Pizzino et al., 2017) In recent years, the importance of OP and ROS in air pollution research has gained increasing attention as these parameters might play a major role in PM-induced diseases. For example, the European parliament recently adopted a revised Ambient Air Quality Directive for Europe.(European Parliament, 2024) It suggests the expansion of monitored pollutants at supersites to also include OP among other pollutants of emerging concern next to the standard network of measurements.(European Union, 2022)

Usually, ROS and OP measurements have been based on the collection of PM filters and subsequent laboratory analysis. However, the time lag between sample collection and analysis can be long, potentially leading to a severe underestimation of



PM OP due to the instability and therefore short lifetime of ROS. Recent studies showed that only a minor fraction (1-40%) of particle-bound ROS in organic aerosol collected on filters is stable on a time scale of up to a week compared to direct-to-

reagent online measurements, emphasizing the need for immediate analysis of particles for accurate quantification of ROS and OP.(Campbell et al., 2023; Zhang et al., 2021) Other studies have also shown short half-lives of compounds contributing to ROS and OP in aerosol particles such as radicals (Campbell et al., 2019), peroxy acids (Steimer et al., 2018), and hydroperoxides (Zhao et al., 2018) ranging from minutes to hours. These findings indicate that online measurement methods that utilise a direct-to-reagent sampling approach are required for robust quantification of OP and ROS, particularly to capture

the short-lived reactive components.

PM originating from anthropogenic sources typically has higher OP than natural emissions, as evidenced by a recent study which observed a factor of three higher OP for anthropogenic compared to biogenic secondary organic aerosol (SOA).(Daellenbach et al., 2020; Zhang et al., 2021) Two important classes of anthropogenic aerosols are emissions from

residential wood combustion (RWC) and traffic emissions. The anthropogenic contribution from RWC is especially visible during winter, when most of $PM_{2.5}$ comes from biomass burning. (Gon et al., 2014) Even at very low ambient concentrations of PM attributed to residential wood combustion, there is an observable health impact. A recent study estimated that annual concentrations of RWC aerosol as low as 0.46 µg/m$^3$ can lead to a decrease in life expectancy of 0.1 years.(Orru et al., 2022) Additionally, emissions from road traffic are a major source of urban air pollution. While modern exhaust treatment systems

can reduce primary emissions of PM, the formation potential of secondary emissions is still high, even with engines complying to the newest regulations. (Gao et al., 2021; Hartikainen et al., 2023; Platt et al., 2017)

Recent studies have attempted to uncover the relationship between composition and potential health effects of PM. Specific components like organic/elemental carbon and metals were associated with an increased risk of negative health outcomes.(Atkinson et al., 2015; Heo et al., 2014) Thus, the composition of aerosol particles and their emission sources play

a key role in dictating OP and ROS formation. Determining the sources and components which influence OP provides crucial information for policymakers to more efficiently abate air pollution.

In this study, the particulate OP and ROS characteristics of a gasoline passenger car as well as a residential wood stove emissions were investigated. Two recently developed online instruments which allow for high time resolution (10 min) OP and ROS measurements were deployed: the online oxidative potential ascorbic acid instrument (OOPAAI) measures aerosol

particle OP using an online ascorbic acid assay as described by Utinger et al., 2023 (Utinger et al., 2023) and the online particle-bound ROS instrument (OPROSI) quantifies particle-bound ROS.(Wragg et al., 2016) Besides sampling the emissions directly they were also passed through a photochemical flow tube reactor to simulate atmospheric ageing. Furthermore, we calculated emission factors for OP ($EF_{OP}$) and ROS ($EF_{ROS}$) potentially allowing to assess health risks associated with exposure to PM emitted from these two sources and to provide information on a potential link between atmospheric ageing of particles and

oxidative stress.



## 2 Material and Methods

### 2.1 Reagents

All chemicals were obtained from Sigma-Aldrich and were used without further purification unless otherwise indicated: ascorbic acid (99.0%), Dehydroascorbic acid (99.0%), 0.1 M HCL, 0.1 M NaOH solution, Chelex 100 sodium form, o-phenylenediamine (≥99.5%), HEPES (4-(2-hydroxyethyl)-1-piperazineethanesulfonic acid ≥ 99%), methanol (99.9%), peroxidase from horseradish (Type VI, HRP), DCFH-DA (2,7-Dichlorofluorescin diacetate, 97%), 3% hydrogen peroxide solution, 1 M PBS solution, zero grade air (Model 737–250, Aadco Instruments Inc., USA), $N_2$ gas (purity 99.999%, Linde, Finland).

### 2.2 Aerosol Generation and Characterization

Here, only a brief overview of the experimental setup of emission generation and ageing, as well as sampling, is given. More details can be found in Mukherjee et al., 2024, and Paul et al., 2024.(Mukherjee et al., 2024; Paul et al., 2024) Figure 1 illustrates a simplified schematic of the experimental setup. The raw exhaust was sampled directly at the tail pipe of a EURO 6d gasoline car (Škoda Scala 2021) or the chimney of a residential wood stove (Aduro 9.3). Flowing through a heated sampling line, a two-step dilution system reduced the concentration and temperature of the sample using a porous tube and an ejection diluter. The dilution ratio was set at 1:17 during the car emission experiments and 1:60 during the RWC experiments because

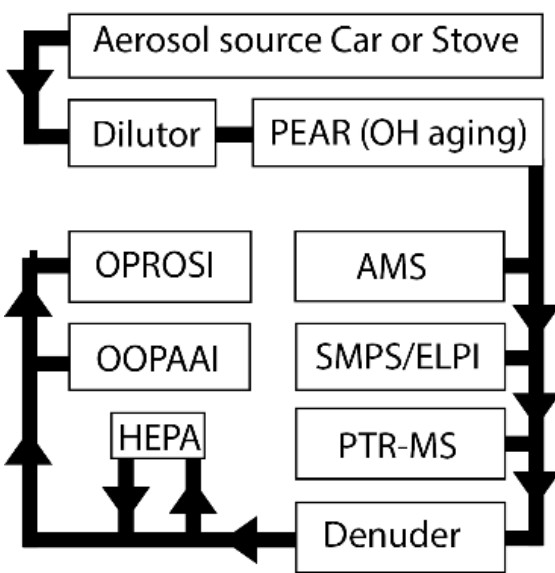

**Figure 1: A simplified schematic of the components and analysis instruments relevant for this paper. The aerosol is transported and diluted from the emission source to the PEAR chamber and after further dilution and removal of reactive gas-phase components with a denuder, the aerosol was characterised for OP and ROS.**



of the large difference in PM and gas phase concentrations of the two PM sources. The exhaust flow then passed through the photochemical emission ageing flow tube reactor (PEAR) to simulate atmospheric ageing.(Ihalainen et al., 2019) For primary experiments the PEAR was not in operation, but the aerosol was passed through it nonetheless to have comparable results. The equivalent photochemical ages were determined to be between 1.1 and 5.1 days using fully deuterated butanol.(Paul et al., 2024; Schneider et al., 2024) The OOPAAI and OPROSI were connected after the PEAR to measure both primary and secondary emissions. Primary emissions without oxidative ageing by the PEAR were measured as well, to allow for the comparison between primary and aged combustion particles. Due to high particle concentrations during the RWC experiments, an additional porous tube diluter was connected and set to a dilution ratio of 1:1.5 to 1:3, depending on experimental conditions. By passing the sample through a high efficiency particle arresting (HEPA) filter (HEPA-CAP150, Whatman), blank measurements were possible during experiments to check for background drifts or gas-phase artefacts. Subsequently, two 1 m long home-built denuders filled with activated charcoal (untreated, granular, Sigma-Aldrich) as well as one denuder with 8 honey-comb shaped charcoal elements (Ionicon) were connected in-line to remove any reactive gas-phase components that could contribute to the online ROS or OP signals (e.g., $O_3$ or oxidizing volatile organic compounds). The charcoal was regenerated at 230 °C for 24 h every second day. Having both instruments connected to the same denuders allowed for a higher sample flow through them, reducing particle losses, which were estimated to be 10% and the data was corrected by that value (Figure S 1).

The car was operated following a driving cycle, where one cycle lasted for an hour and an experiment consisted of four cycles equalling to 4-hour measurement periods. Every cycle consisted of four different steady-state driving conditions starting with 5 min of idling, followed by each 15 min of 50 km/h (4th gear), 100 km/h (5th gear), 80 km/h (5th gear) and ending with 10 minutes of idling. For the RWC measurements, an initial batch of beech wood logs (1.85 kg) with 150 g of beech kindling was ignited. Similar procedures were used in other studies using the same stove.(Ihantola et al., 2022; Leskinen et al., 2023; Martens et al., 2021) After every 35 min an additional batch of 2 kg of beech logs was added for a total of six batches. After the last addition, the wood was left to burn out (ember phase) for 30 min during which the supply of fresh air was stopped. One RWC experiment lasted 4 h. Blank OP and ROS measurements were performed before and after experiments as well as at different time points during an experiment to characterize potential gas-phase contributions, denuder efficiency, and instrument backgrounds. The OOPAAI was started the day before an experiment day and blank measurements were run overnight. The OPROSI was started at least one hour before the start of an experiment. The filters inside the liquid systems of both instruments were changed daily to avoid excessive contamination by insoluble particles. Additionally, this was done between each 4 h run of the RWC experiments for the OPROSI. The OOPAAI and OPROSI were calibrated once a week to ensure consistent performance for the duration of the measurement campaign. Otherwise, they were operated as described in Wragg et al., 2016 (Wragg et al., 2016) and Utinger et al., 2023.(Utinger et al., 2023)



### 2.3 Chemical Preparation

All chemical solutions used in this study were made immediately prior to measurements, except where otherwise specified. Water used to prepare the solutions was purified by a high purification water unit (resistivity 18.2 MΩ/cm). To further reduce the amount of contamination, water was passed through a fritted column filled with 100 g Chelex 100 resin. The valve was adjusted to a flow of one drop per minute through the resin. The treatment was used to remove trace metals (i.e. copper and iron) to avoid interference with the AA oxidation caused by the sample. A 200 mM HEPES buffer stock solution to adjust the pH to a physiological relevant range (pH 6.8) was prepared monthly using Chelex treated water and stored at 4 °C. HEPES was used because it has a lower chelating effect of transition metals than PBS.(Utinger et al., 2023) AA solutions were prepared the day before the experiments in an effort to stabilize the background drift caused by the autooxidation of AA. 8.8 mg of AA were dissolved in 25 mL HEPES stock solution and 225 mL Chelex treated water to form a 200 μM AA working solution. OPDA solution was prepared by dissolving 0.54 g in 250 mL of 0.1 M hydrochloric acid to form a 20 mM working solution. HRP stock solution was prepared weekly by dissolving 5000 units in 500 mL of water and stored at 4 °C. HRP working solution was prepared using 100 mL of HRP stock solution, 100 mL of 1 M PBS, and 800 mL of pure water. DCFH-DA stock solution was prepared weekly by dissolving 50 mg of DCFH-DA in 50 mL of methanol and sonicating it for 15 s and stored at −20 °C. DCFH working solution was prepared by reacting 4 mL 0.1 M NaOH with 4.872 mL of the DCFH-DA stock solution for 30 min in the dark. Subsequently, 100 mL of 1 M PBS and pure water were added to a total volume of 1 L. The HRP and DCFH working solutions were prepared the night before and stored at 4 °C. For both assays a calibration was performed once per week as described in Utinger et al., 2023 for ascorbic acid and Wragg et al., 2016 for DCFH to convert the raw fluorescence signal to the corresponding equivalents using known concentrations of DHA and $H_2O_2$. (Utinger et al., 2023; Wragg et al., 2016)

### 2.4 Data Analysis

The fluorescence spectroscopy data was first corrected for the blank background and its drift over the four-hour measurement period. This was done by applying a fit across all blank measurements and subtracting the fitted blank values from the raw signal. The drift was less pronounced in the OOPAAI for the car exhaust measurements and a linear fit was sufficient. However, for the stronger drifts during the RWC measurements, the baseline was fitted using a B-spline function in OriginPRO (OriginLab 2023). For the OPROSI corrections, a polynomial fit of 3$^{rd}$ order was used for the car and RWC data. The data were averaged to 10 s to match the electrical low-pressure impactor mass measurements (ELPI, Dekati) providing total $PM_{2.5}$ mass concentrations. The ELPI particle size range used for the car measurements was $PM_{0.3}$ while $PM_{0.6}$ was used for the RWC emissions. These size ranges were chosen because gaseous ions in the exhaust gas would lead to measurement artefacts and a considerable overestimation of total PM mass in the larger size bins of the ELPI. This can be shown as no significant particle concentrations were present above these size ranges as measured by the SMPS as well as ELPI (Figure S 3). (Paul et al., 2024) Taking these ELPI size bins, the mass concentrations were in a similar range as measured by the SMPS whiles still offering a





higher time resolution (Figure S 4). A constant particle density of 1.6 g/cm³ was used for mass calculations of the car
measurements (Paul et al., 2024) and for PM emitted by the stove densities from Mukherjee et al., 2024, were used (Mukherjee
et al., 2024). Average values of PM densities of 1.5, 1.75, and 1.85 g/cm³ during the flaming and residual burning phases were
used for fresh, short aged, and medium-aged measurements, respectively. The methods to derive ROS and OP emission factors
(EF) for car exhaust and for RWC PM are described in Paul et al., 2024 and Reda et al., 2015, respectively. (Paul et al., 2024;
Reda et al., 2015)

## 3    Results and Discussion


OP and ROS concentrations of tail pipe particle emissions from a EURO 6d gasoline car as well as from beech wood
combustion using a residential wood stove were quantified. Fresh emissions were characterised but also changes in chemical
properties were studied caused by photochemical ageing in a flow tube reactor. The OP and ROS results are presented
normalized to the sampled air volume ($OP_V$ and $ROS_V$) as well as normalized to PM mass ($OP_M$ and $ROS_M$).

### 3.1  OP and ROS concentrations of Primary and Secondary Car Exhaust Emissions


Four-hour experiments were performed and consisted of four one hour driving cycles with an idling phase and three engine
loads as described in the method section.

Both instruments could not detect any signal for the primary exhaust (Figure S 5). As part of the EURO 6d regulations, the
car's exhaust system is fitted with a gasoline particle filter (GPF) reducing primary exhaust emissions.(Paul et al., 2024) Thus,
possible particle OP and ROS activity in primary particulate emission were below the detection limit of our online instruments.
In contrast, as illustrated in Figure 2, significant $OP_V$ and $ROS_V$ concentrations and particle mass ($PM_{2.5}$) were measured for
photochemically aged car exhaust for an equivalent atmospheric ageing of 2.1 days in the PEAR chamber with OP and ROS
concentrations of up to 800 nmol DHA m⁻³ and 2200 nmol $H_2O_2$ equivalents m⁻³ were measured. Aged particle emissions
consisted only of secondary organic and inorganic particles. Figure 2 clearly demonstrates that both instruments are sensitive
to SOA, as also shown in previous studies (Utinger et al., 2023; Wragg et al., 2016) while inorganic secondary particles, mainly
composed of $NH_4$, $SO_4$ and $NO_3$, do not react with the AA and DCFH in our online instruments.(Paul et al., 2024) $OP_V$ and
$ROS_V$ concentrations generally follow the particle mass closely, although some changes in PM mass are not reflected in $OP_V$
and $ROS_V$. This shows that different engine loads not only cause a change in total aged PM mass concentration, but also in
particle OP and ROS activity. During the last cycle (Figure 2, after 03:00), blank OP and ROS measurements were performed
by removing all particles from the exhaust flow with a HEPA filter in front of the OOPAAI and OPROSI. These blank
measurements clearly illustrate that gaseous components in the aged exhaust were removed efficiently by the charcoal denuders

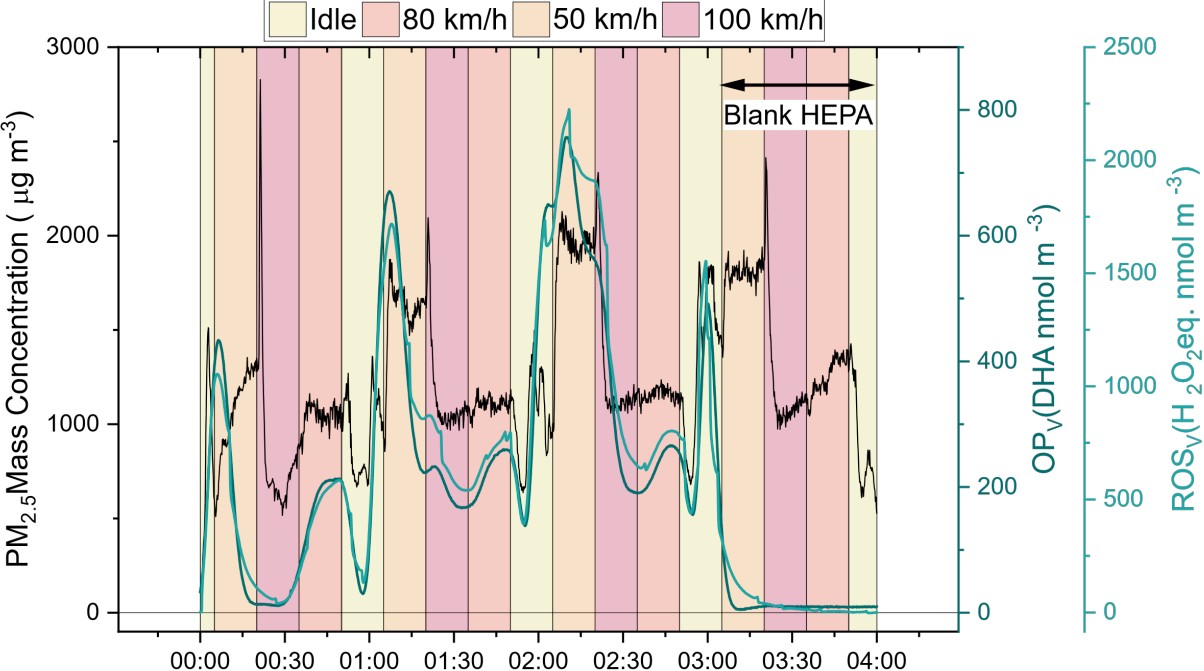

**Figure 2: $OP_V$ and $ROS_V$ measurement of secondary car emissions after an equivalent of 2.1 days aging during four driving cycles as well as the particle mass measured by an ELPI. $OP_V$ and $ROS_V$ are given in nmol DHA/m$^3$ and nmol $H_2O_2$ eq./m$^3$, respectively. The last hour of measurements after 03:00 was used for a HEPA blank, removing all particles from the sample flow.**

in our instruments (Figure 1) and do not cause any measurement artefacts, since both instruments returned to blank-level values.

Two different ageing conditions were applied to investigate the potential influence of photochemical ageing on the intrinsic OP and ROS. In Figure 3 the averaged $OP_M$ of the two ageing conditions are shown. Photochemical ages of approximately 2.1 days (Figure 3A) and 5.1 days (Figure 3B) represent short and medium ageing processes in the atmosphere. Overall, the $OP_M$

was similar between the two ageing times. We observed about two times higher peak $OP_M$ concentrations during idling and at the beginning of the 50 km/h period compared to the other engine conditions with 100 and 80 km/h loads during the short ageing condition ($0.31 \pm 0.19$ nmol DHA µg$^{-1}$ vs. $0.15 \pm 0.03$ nmol DHA µg$^{-1}$). This difference was larger by about 40% during the medium ageing condition ($0.41 \pm 0.29$ nmol DHA µg$^{-1}$ vs. $0.12 \pm 0.08$ nmol DHA µg$^{-1}$). The idling and 50 km/h periods also show the highest variability across all averaged 15 driving cycles due to fluctuations of the emissions. Interestingly, a

consistent short increase in $OP_M$ of 65% at the beginning of the 100 km/h condition was measured (at about 00:25). This could be due to the strong acceleration of the engine similarly to the transition from idle to 50 km/h, causing non-ideal combustion conditions.








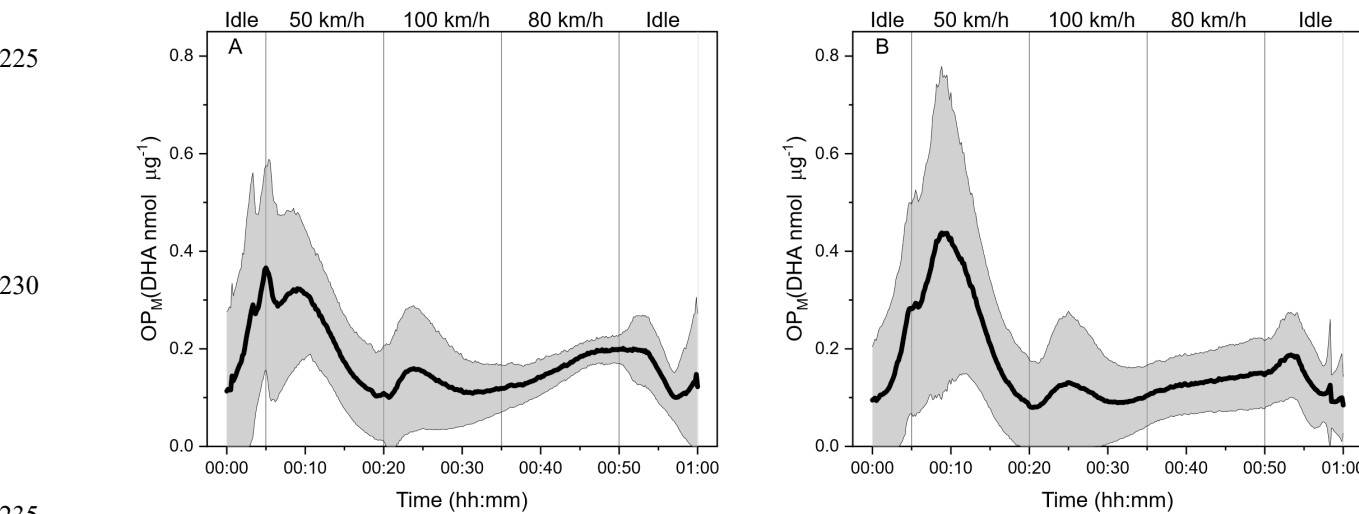

**Figure 3: OP$_M$ concentrations after 2.1 days (A) and 5.1 days (B) photochemical ageing of car emissions. The average of 15 driving cycles is represented in one cycle (black line). The error band (grey shaded area) shown is the standard deviation of the 15 driving cycles.**

The ROS$_M$ concentrations of the two photochemical ageing conditions are shown in Figure 4. Overall, ROS behaviour was similar to the OP$_M$ results across all driving conditions with 40% higher ROS$_M$ values observed during the transition from idle to 50 km/h compared the concentrations measured during 100 and 80 km/h periods and the same slight and transient increase of approximately 30% at the beginning of the 100 km/h (00:25) condition was observed. Unlike the OP$_M$, the ROS$_M$ signal decreased with aging and was 30-70% higher during short ageing (Figure 4A) compared to the medium ageing (Figure 4B)

across all driving conditions. A possible explanation for this decrease of ROS with increasing ageing time could be photolytic decomposition of peroxides over time. Other studies using an atmospheric simulation chamber also observed a decrease in ROS activity with longer photochemical ages (several hours to days) for two-stroke scooter engine emissions as well as biogenic SOA.(Epstein et al., 2014; Platt et al., 2014)

The OP and ROS activity during cold starts at the beginning of an experiment is not distinguishable from following driving

cycles during a four-hour experiment. Because this period is very short (5 min) and thus within the time resolution of both the OOPAAI and OPROSI instruments the potential effect would be difficult to resolve.




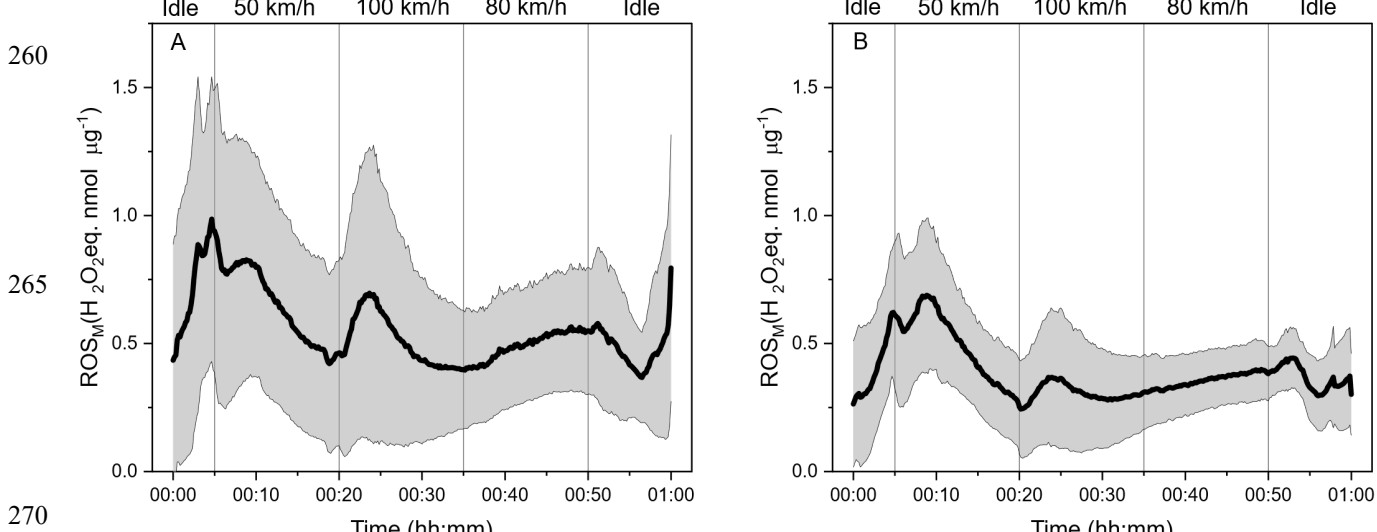

**Figure 4: ROS$_M$ concentration of 2.1 days (A) and 5.1 days (B) in aged car emissions. The average of 15 driving cycles for short ageing and medium ageing is represented in one cycle (black line). The error band (grey shaded area) shown is the standard deviation of the 15 driving cycles.**

## 3.2 OP and ROS Concentrations from Residential Wood Combustion (RWC) Particles

For the RWC experiments, primary emissions as well as two ageing conditions were investigated. In Figure 5 OP$_V$ and ROS$_V$ measurements are shown as well as the particle mass measured by an ELPI during a four-hour experiment with RWC aged for an equivalent of up to 3.3 days. Light and darker blue background colours indicate the addition of a new wood batch (see method section). The last batch was left to burn for additional 30 min as ember phase (yellow background).

Both instruments observed an increase and subsequently a decrease in OP$_V$ and ROS$_V$, respectively, with each batch. OP$_V$ and ROS$_V$ reached peak values in the first half of a 35 min long batch with approximately a factor of ten higher compared to values at the end of a batch. This again demonstrates how both online instruments are capable of capturing fast changing emission characteristics. After six batches of RWC the ember phase started, characterised by low particulate emissions. The OP$_V$ and ROS$_V$ signals gradually decreased accordingly. Compared to the car experiments, the RWC results showed a higher batch-to-batch variability, observed by both instruments as well as the particle mass concentration which was also reported in previous RWC studies.(Heringa et al., 2012; Vicente et al., 2015)



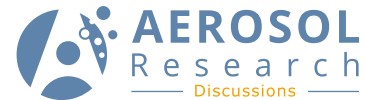

In contrast to the car emissions, both instruments responded to the primary RWC emissions due to the significantly higher primary PM emissions. The OOPAAI observed an OP signal for primary emission which was up to three times lower than during the aged experiments (6), which can be expected as ROS are predominantly formed through oxidation processes.

In contrast, the OPROSI showed a different response to primary RWC particles with negative ROS values (Figure S 6), which

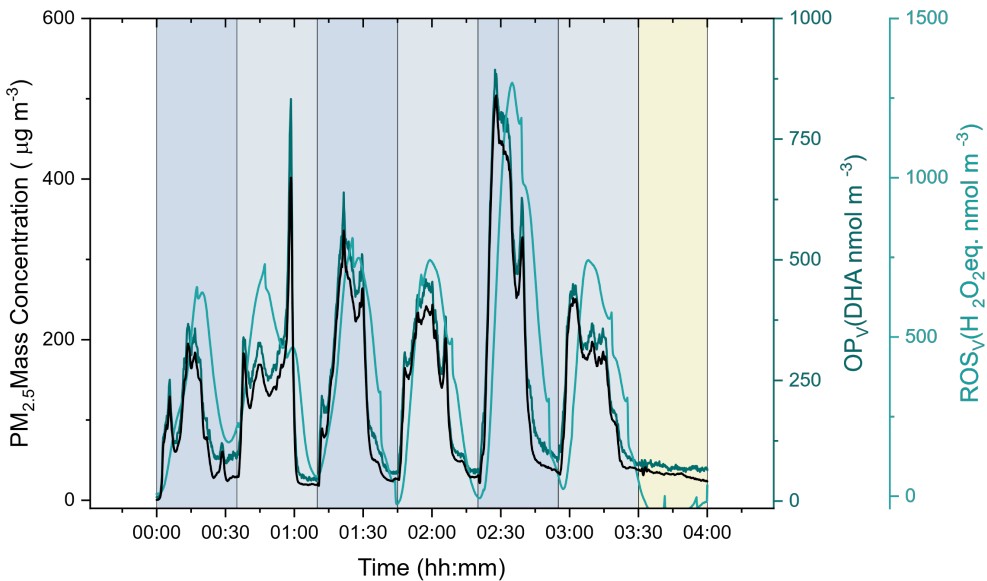

**Figure 5: OP$_V$ (dark green) and ROS$_V$ (light green) measurement of one secondary RWC emissions experiment as well as the particle mass (left axis, black line) measured by an ELPI. The blue boxes mark the different batches of wood added to the oven and the yellow box the ember phase.**

might be a measurement artefact of the large fraction of insoluble primary PM (e.g. soot). Due to the high PM and specifically soot concentrations during these emission measurements, particles accumulated inside the instruments and potentially interacted with and deactivated the assay, causing the ROS signal to fall below blank values. The hydrophobic and large surface area of the soot particles could adsorb and inactivate reagents of the assay (e.g. HRP), causing an apparent lower ROS
concentration compared to the blank. Enzymes are known to change their activity due to absorption onto a high surface area substrate.(Khan, 2021) To test this hypothesis in a qualitative way, activated charcoal was mixed with the ROS assay as a proxy for soot since it has also a high hydrophobic surface area. Figure S 2 shows a clear decrease of the ROS signal with higher concentrations of charcoal. Organic components in the primary particles, e.g. antioxidant compounds frequently found in wood smoke such as phenols, could also cause a signal decrease by reacting with HRP.(Kjällstrand and Petersson, 2001)
The negative ROS peaks were less than 10% of the positive signals for the aged exhaust and thus any uncertainties of aged ROS values related to the very high soot or antioxidant conditions are likely in the same range.



Photochemical ageing led to a significant increase of the $OP_M$ and $ROS_M$ activity compared to fresh emissions (Figure 6), while the PM mass range stayed within the same order of magnitude as during the primary measurements. In Figure 6, the $OP_M$ of the primary aerosol and the two ageing conditions are shown with the standard deviation from 16-21 repeat
measurements. Two different ageing conditions with $1.4 \pm 0.2$ (short) and $3.3 \pm 0.4$ days (medium) were investigated. The ember phase is not considered in these figures since it did not show any detectable OP or ROS activity (data not shown) The primary aerosol (Figure 6A) resulted in low and stable $OP_M$ values in the same range as the car emissions for the majority of a batch. Near the end, when the wood was almost burned up and when no flames were visible anymore in some cases, $OP_M$ increased sharply by up to a factor of 6 compared to the rest of the batch. A continuous increase was observed during the short
ageing condition with peak values being reached in the last 10 minutes of a batch (Figure 6B). The average $OP_M$ values remained comparable to the primary aerosol during the same time period within a large batch-to-batch variability ($5.19 \pm 9.48$ nmol DHA $\mu g^{-1}$ vs. $6.48 \pm 8.30$ nmol DHA $\mu g^{-1}$). The large increase in $OP_M$ at the end of a batch (as seen for primary emissions and short aging) was no longer observed while $OP_M$ concentrations still increased continuously during the course of a batch, similar to short aging conditions (Figure 6C). This resulted in an overall lower $OP_M$ with medium ageing.

The reduction of $OP_M$ from short to medium ageing could be due to chemical changes of PM components caused by prolonged oxidation reactions. For example, the oxidation of PAHs in the atmosphere, formed during wood combustion, can lead to the formation of quinone-type products, (Walgraeve et al., 2010) which are known to produce ROS in an aqueous solution and therefore would also contribute to OP in the OOPAAI.(Charrier and Anastasio, 2012; Li et al., 2003; Njus et al., 2023).

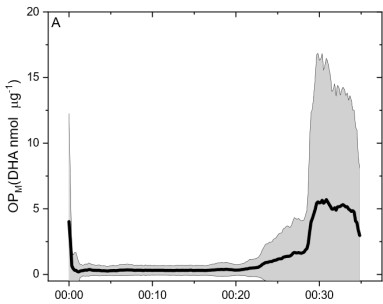 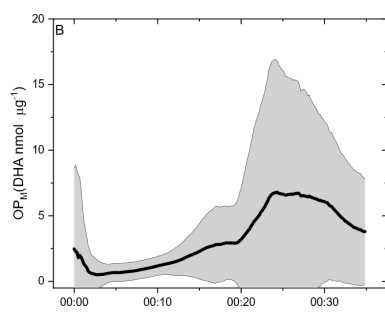 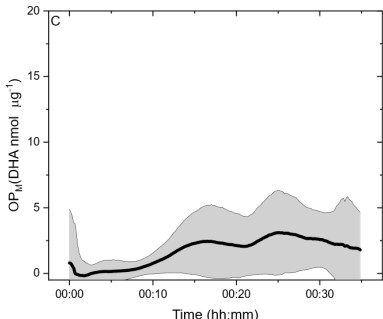

**Figure 6: Average $OP_M$ of  0 days (fresh, 21 batches, A), $1.4 \pm 0.2$ days (short, 16 batches, B) and $3.3 \pm 0.4$ days (medium, 17 batches C) RWC emissions (black line). The error showed is the standard deviation of the averaged batches (grey shaded area).**

However, continued oxidation during medium ageing may result in further oxidation of such OP-active quinones into inactive
compounds, which would result in a lower OP activity. This effect has been observed for markers of biomass-burning SOA before through the breakdown of aromatic rings.(Fang et al., 2024) Wong et al. observed an overall decrease of OP in aged laboratory-generated biomass burning aerosol after a short initial increase similar to our OP measurements with ascorbic acid.(Wong et al., 2019) This overall decrease with higher ageing could also partially be explained by the decrease in quinone concentrations in wood smoke observed with higher ages.(Jiang and Jang, 2018) Also for certain SOA types from gaseous



precursors β-pinene and naphthalene, it was observed that more ageing does not induce a higher OP.(Offer et al., 2022) Figure
      7 shows the $ROS_M$ values for the two ageing conditions. For the primary RWC aerosol ROS signals could not be determined
      as discussed above. Similar to $OP_M$, the highest $ROS_M$ values were observed in later parts of a batch during both ageing
      conditions. However, this trend was more pronounced during the medium ageing experiments, in contrast to $OP_M$. $ROS_M$
      during short ageing (Figure 7A) was three times lower compared to the medium ageing results (Figure 7B). More oxidized
aerosol is generally associated with more oxygenated fraction of PM such as peroxides. To which the DCFH assay is especially
      sensitive.(Fang et al., 2024; Li et al., 2021; Nordin et al., 2015; Wang et al., 2023; Zhang et al., 2021) This is in agreement
      with aging of wood smoke in an atmospheric simulation chamber, where ageing by OH also lead to an increase in ROS
      activity.(Wang et al., 2023) Zhang et al. (2021) also measured an increase in ROS with higher ageing times for two simple
      SOA systems using β-pinene and naphthalene as precursors.(Zhang et al., 2021)


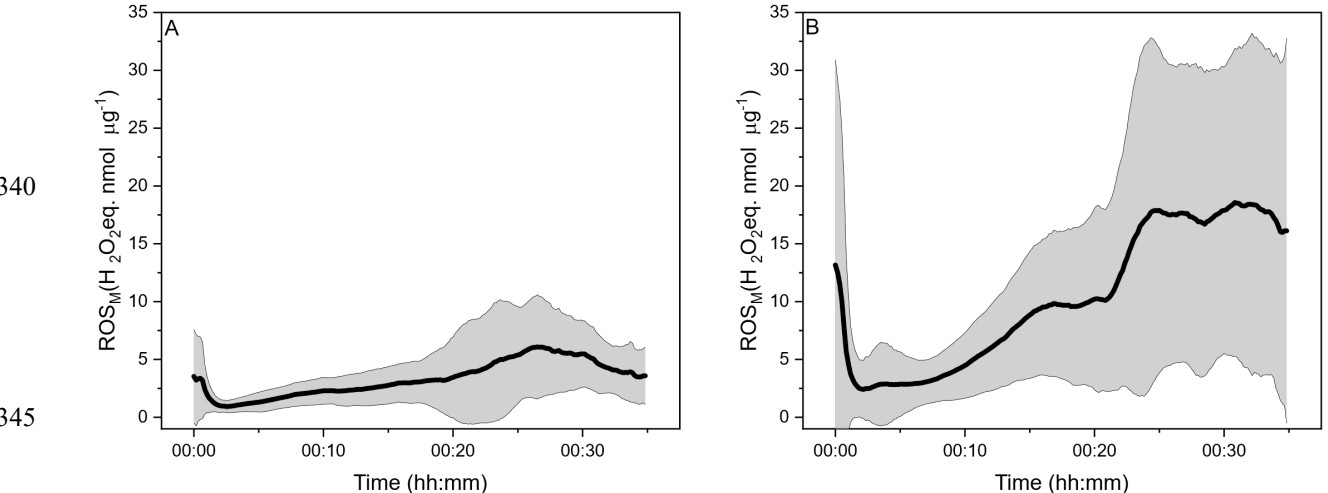

**Figure 7: Average $ROS_M$ concentrations for 1.4 ± 0.2 days (short, B, 16 batches) and 3.3 ± 0.4 days (medium, C, 17 batches) RWC emissions (black line). The standard deviation of the different measurements is plotted as an error band (grey shaded area).**


### 3.3  Comparison of $OP_M$ and $ROS_M$ and Emission Factors of Car Exhaust and RWC

Table 1 summarizes the $OP_M$ and $ROS_M$ values we observed covering a realistic range of atmospheric oxidative processing times up to 5 days. The $OP_M$ and $ROS_M$ activity measured during this study were highly dynamic, changing up to two orders of magnitude within minutes and are comparable to other lab studies characterising pure SOA systems conducted with similar
online instruments.(Campbell et al., 2023; Zhang et al., 2021)



**Table 1: Overview of the measured $OP_M$ and $ROS_M$ values illustrating the large range observed within a cycle (car) or batch (RWC) as a result of the high-time-resolution online measurements. To determine the atmospheric age (in days) for the RWC measurements the average and standard deviation of 5 ageing condition measurements is given, while for the car exhaust only one measurement per age was conducted.**

| Source | Age (days) | $OP_M$ (nmol DHA $\mu g^{-1}$) | $ROS_M$ (nmol $H_2O_2$ eq. $\mu g^{-1}$) |
|--------|-----------|-------------------------------|------------------------------------------|
| Car | 0.0 | N/A | N/A |
| | 2.1 | 0.1-0.4 | 0.4-1.0 |
| | 5.1 | 0.1-0.4 | 0.2-0.7 |
| RWC | 0.0 | 0.2-5.7 | N/A |
| | 1.4 ± 0.2 | 0.5-6.8 | 0.9-6.1 |
| | 3.3 ± 0.4 | 0.0-3.1 | 2.4-18.6 |

A comparison of primary particles is not possible, because the car has a GPF and therefore almost no primary particle emissions, in contrast to the stove with significantly more primary particles. This means that during the stove experiments combined effects of primary and secondary aerosols on the OP and ROS activity were observed while only secondary car particles were measured. The lack of primary particles could partially be responsible for the up to one order of magnitude higher $OP_M$ and $ROS_M$ values from the wood stove. The large difference in $OP_M$ and $ROS_M$ between car and RWC emissions could also be explained by the difference in composition of the secondary particles A significant compositional difference is the metal content of the two aerosol types. Wood smoke is known to contain a wide range of redox active metals, including zinc, iron and copper.(Erlandsson et al., 2020; Gonçalves et al., 2010; Uski et al., 2015) The total metal concentrations can reach up to 2.5 wt.% of $PM_{2.5}$ depending on wood type and combustion appliance whereas car exhaust treated with a particulate filter only contains very low concentration of metallic primary particles and consists mostly of secondary aerosol.(Alves et al., 2011) The presence of transition metals has been shown to have an influence on the OP and ROS activity of SOA. Campbell et al. observed synergistic effects leading to higher $OP_M$ when biogenic SOA and metals are combined.(Campbell et al., 2023) In addition, differences in SOA composition from these two sources are likely also contributing to the observed differences in $OP_M$ and $ROS_M$.

Emission factors are calculated and used to estimate the quantity of pollutants released into the atmosphere from various sources, helping to assess environmental impacts and guide regulatory compliance and mitigation efforts. To the best of our knowledge, emission factors of OP and ROS have never been reported in the literature before. Figure 8A shows $EF_{OP}$ and $EF_{ROS}$ for different driving speeds and both ageing conditions in pmol DHA/kg fuel and pmol $H_2O_2$ eq./kg fuel, respectively. Similar to the PM mass normalized OP and ROS results, the 50 km/h condition resulted in the highest EF values. Medium ageing led to an overall increase of $EF_{OP}$ and $EF_{ROS}$. The smallest change was observed for a speed of 100 km/h. Figure 8B shows emission factors for RWC for fresh emissions, as well as short and medium ageing. Opposite effects of ageing on the two assays are visible similar to trends in the mass-normalised $OP_M$ and $ROS_M$ data (Fig. 6 and 7): An increase in ageing leads to a reduction of $EF_{OP}$ but to an increase of $EF_{ROS}$. $EF_{OP}$ of both aging conditions (blue and blue stripes for the car and short



and medium for RWC) are very similar for both sources. For $EF_{ROS}$ and $EF_{OP}$ gasoline car emissions are up to 8 times higher

than RWC values, which is the opposite compared to mass normalised $OP_M$ and $ROS_M$ concentrations. $EF_{OP}$ and $EF_{ROS}$ could

be a useful metric to compare OP and ROS per kilogram of fuel to assess the relative toxicity of emissions from different

combustion sources.


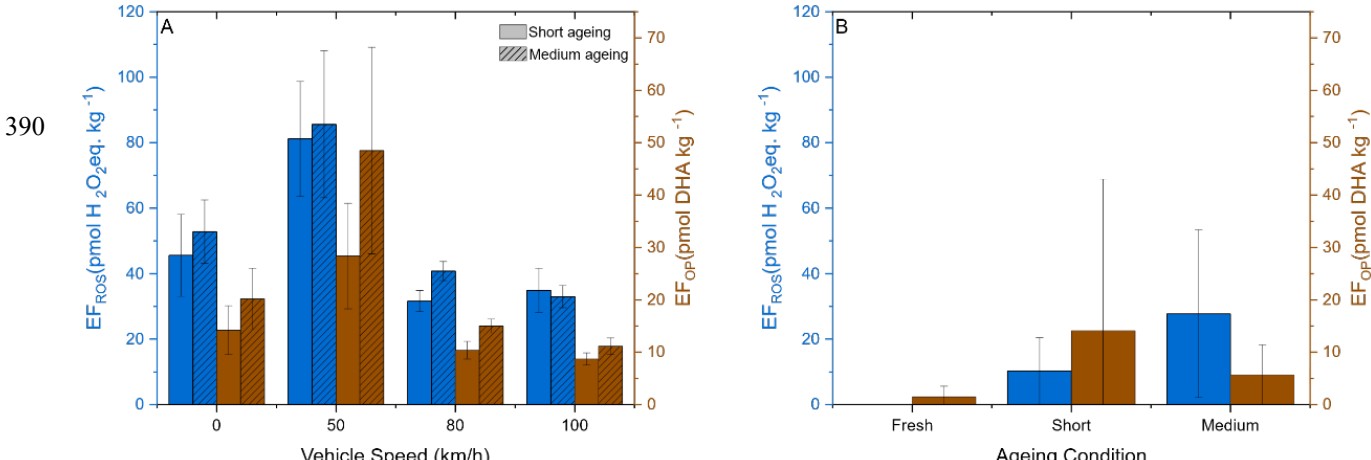

**Figure 8 Emission factors for car emissions (A) and RWC (B). The blue bars represent $EF_{ROS}$ while $EF_{OP}$ is shown in brown. The striped bars represent the medium ageing condition whereas the plain bars show short ageing. The error bars shown represent the standard deviation of the averaged values and indicate the variability of the measurements.**

## 4   Conclusion

OP and ROS concentrations of aged and fresh emissions from a gasoline car as well as wood burning stove were characterized

with two online, high-time resolution instruments. On average, the RWC emissions had a up to ten times higher $OP_M$ and

$ROS_M$ than the car emissions. Atmospheric ageing affects the $OP_M$ and $ROS_M$ content in particle emissions of these two sources

differently: for RWC an increased $OP_M$ was observed with short ageing compared to the fresh emissions and medium aging

but for car exhaust, longer ageing leads to a slightly higher $OP_M$.

In contrast, the $ROS_M$ content decreased with higher ageing for the car emissions. For the RWC emissions the opposite was

observed: the $ROS_M$ concentration increased with longer ageing.

The highly dynamic changes in OP and ROS activity within one car driving cycle or wood batch, as well as the large batch-

to-batch variability of values (often on time scales of a few minutes), would not be detectable with traditional offline methods.

Our data indicate that the contribution of RWC emissions per µg particle emissions towards PM toxicity is significantly larger

than from gasoline car emissions. However, this is also due to the implementation of a GPF, preventing most primary car

emissions.



Biomass burning is the main emission source from the residential sector which can contribute more than 50% of anthropogenic PM$_{2.5}$ emissions in some parts of Europe.(Zauli-Sajani et al., 2024) The PM$_{2.5}$ mass contribution of biomass burning was shown to be as high as from traffic exhaust even at traffic sites in five European cities.(Saraga et al., 2021) Combined with the high OP and ROS activity of RWC emissions, the potential detrimental effects on human health should be considered in air quality efforts. The potential significance of car emissions should be taken into consideration as well since the SOA particle formation

potential is high, although there are almost no primary particles. These insights should play a major role in air quality guideline evaluations. Overall, the results presented here add to the existing weight of evidence that OP measurements should be considered as a valid metric to evaluate the potential health effects of PM.

## 5  Data Availability

All data can be accessed from the corresponding author upon request.

## 6  Author Contribution

BU, AB writing – original draft preparation, investigation, methodology, formal analysis and visualization, AP, AM, C-MM, MI, PYP, MKortelainen, PM, MS, JL investigation, SJC, HC, OS, writing – reviewing & editing and conceptualization, TH, YR, RZ Resources, MKalberer writing, reviewing & editing, conceptualization, supervision and resources

## 7  Competing Interests

The authors declare that they have no conflict of interest.

## 8  Acknowledgments

We thank the SNF (Grant 192192), the Helmholtz international Laboratory aeroHEALTH (InterLabs-0005; https://aerohealth.eu) and the ULTRHAS Horizon Europe project (agreement 955390) for supporting this work and every one

of the aeroHEALTH consortium that contributed to the measurement campaign.





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
