# Peer review of "Emission dynamics of reactive oxygen species and oxidative potential in particles from a gasoline car and wood stove"

_Aerosol Research, 2024_

## Author Comment (AC1)

**Answer Reviewer 1**

The study investigates the effects of chemical aging on the formation of reactive oxygen species (ROS) and oxidative potential (OP) of aerosol particles emitted from residential wood burning and car exhaust using semi real-time instruments. Overall, the study is designed well and the results provide some insights into the dynamics of OP and ROS evolution during chemical aging compared to primary emissions. However, the manuscript text and the communication of study design and results require improvements before the manuscript can be accepted for publication. I have provided some general and specific comments below:

The authors like to thank the reviewer for their constructive comments. Our answers to their questions and comments are given below in blue font.

**General comments:**

The objectives of the study have not been defined well. The authors state the work they carried out without saying why and how this study is different from their previous works using the same instruments. What this study aims to add to the existing literature in the field? The objectives need to be itemized at the end of the introduction section and correspond to sub-sections in the results section. We changed the last paragraph of the introduction of the manuscript according to the reviewer's recommendation (line 96-106):

"In this study we address this knowledge gap and quantify for the first time the highly dynamic particulate OP and ROS characteristics of emissions of gasoline passenger car as well as of a residential wood stove. Two recently developed online instruments were used in this study, which allow for unprecedented high time resolution (10 min) OP and ROS measurements: the online oxidative potential ascorbic acid instrument (OOPAAI) measures aerosol particle OP using an online ascorbic acid assay (Utinger et al., 2023) and the online particle-bound ROS instrument (OPROSI) quantifies particle-bound ROS.(Wragg et al., 2016) Besides sampling the emissions directly (i.e. primary emissions) they were also passed through a photochemical flow tube reactor to simulate two different atmospheric ageing conditions. Combustion conditions have highly dynamic emissions profiles and only online instruments, as used in this study, are able to characterise fast-changing OP- and ROS properties of particle emissions. Furthermore, we calculated for the first time emission factors for OP ($EF_{OP}$) and ROS ($EF_{ROS}$) enabling the assessment of health risks associated with exposure to PM emitted from these two sources and to provide information on a potential link between atmospheric ageing of particles and oxidative stress.."

The discussions appear rather qualitative often without mentioning the measured quantities and how they compare in numerical terms. Moreover, statistical analysis is completely missing when describing the results. Considering that the study compares various experimental conditions and how they affect ROS and OP measurements, statistical analysis should be included and differences in results should be stated with statistical significance. For instance, considering the large variabilities in Figure 4, is ROSm "significantly" different in primary vs. aged emissions during 100 km/h cycle, or with primary emissions during 50 vs. 80 km/h cycle? This comment applies to other (similar) cases throughout the text.

We agree with the reviewer that a more detailed statistical analysis would be desirable. However, the highly dynamic nature of ROS and OP emissions during the driving cycles for the car and the different batches during the wood burning experiments did unfortunately not allow to determine statistically significant differences in OP and ROS emissions at different speeds (Fig. 3, 4 and 6). In addition, the large variability and therefore standard deviation of OP and ROS are due to the high variability of emissions from these sources, even under controlled lab conditions as in this current study. Other instruments (ELPI, SMPS (see SI Fig 3) characterizing the wood stove PM emission as well as Aerosol Mass Spectrometer analysis (see Paul et al., 2024) of the car show as well a high variability between repeat measurements. To indicate the lack of statistical significance between different driving conditions, we replaced throughout the text all mentioning of "significantly different" etc.  with "trend" etc. to clarify that we can only see trends of the different conditions without statistical significance.

The method section does not discuss the use of aerosol measurement instruments that are shown in Figure 1, in particular AMS and PTR-MS. What were these used for? No data has been presented either. If they were part of this study, they should be clearly mentioned in the method section (with additional details provided in the supplement) and how the related data was used.

The PTR-MS was used to calculate the OH aging time using the butanol reaction kinetics. This method is described in another paper in detail and we cited this publication in line 132.

We describe this method now in more detail (line 129-135): "The equivalent photochemical ages were determined to be between 1.1 and 5.1 days by determining the decay kinetics of fully deuterated butanol measured by a proton-transfer-reaction time-of-flight mass spectrometer (PTR-TOFMS 8000, Iconic).(Paul et al., 2024; Schneider et al., 2024). Furthermore, a scanning mobility particle sizer (SMPS, Model 3776 CPC, Model 3080 classifier) and an low-pressure impactor mass measurements (ELPI, Dekati) were used to quantify the particle number size distributions. The OOPAAI and OPROSI were connected after the PEAR to measure OP and ROS from both primary and secondary emissions"

The AMS was not used for this study and was therefore deleted in in Figure 1.

\*\*\*\*\*\*\*\*\*\*\*\*\*\*\*\*\*\*\*\*\*\*\*\*\*\*\*\*\*\*\*\*\*\*\*\*\*\*\*\*\*\*\*\*\*\*\*\*\*\*\*\*\*\*\*\*\*\*\*\*\*\*\*\*\*\*\*\*\*\*\*\*\*\*\*\*\*\*\*\*\*\*\*\*\*\*\*\*\*\*\*
\*\*\*\*\*\*\*\*\*\*\*\*\*\*\*\*\*\*

**More specific comments:**

Page 2, Line 55-58: a more inclusive language should be used here. Exogenous reactive species exists in the gas phase too and this aspect has been neglected in this study. A sentence has been added to clarify that (line 58-59):

"Exogenous reactive species exist also in the gas phase, but in this study we focus on the particle exposure only."

Page 3, Line 87-91: the paragraph needs to be revised; the association of PM chemical composition, OP, and health effects is not highlighted well. This aspect has been studies in the past and association was found. The authors should clearly describe such links and provide appropriate citation.

We changed the paragraph and added citations (line 88-93):

"Recent studies have attempted to uncover the relationship between composition and potential health effects of PM.(Campbell et al., 2024) OP has emerged as biologically significant chemical metric that may serve as a critical connection between the chemical composition of particles and their associated adverse health effects.(Bates et al., 2019) Specific components like organic/elemental carbon and metals were associated with an increased risk of negative health outcomes.(Atkinson et al., 2015; Heo et al., 2014) Thus, the composition of aerosol particles and their emission sources play a key role in dictating OP and ROS formation."

Page 3, Line 92-100: This paragraph needs to be revised: the authors should (a) clearly indicate the gap in knowledge this study is addressing, and (b) define the specific objective/s it aims to achieve.

This has already been addressed in our reply to the first comment of the reviewer above.

Page 4, Section 2.2.: the text should be re-arranged to follow the order of flowchart/Figure 1 (without skipping any step, as is the case now). Importantly, as mentioned in my general comments, the authors should provide more details about the aerosol measurement instrumentation used in this study (specially AMS, SMPS, and PTR-MS measurements, which appear to be positioned before OOPAAI and OPROSI). Details should be included in the supplement and referenced in the main text. Referencing previous publications is not sufficient.

We added this information in the section 2.2 (line 126-134). References, specifically the description of the PEAR chamber (Ihalainen et al., 2019), and manufacturer of all commercial instruments are given now in the revised text. SMPS and PTR-MS are standard commercial instruments and we think they do not need to be described in detail here.

Page 5, Line 116: provide more information (numerical values) about "the large differences in PM and gas phase concentrations" to make it clear why such different dilution ratios were chosen. The use of qualitative terms should be avoided; there are many such instances in this manuscript.

The different exhaust dilutions (1:17 and 1:60 as mentioned in line 138) were applied to provide particle concentration within the same order of magnitude to the different analysis instruments assuring consistent instrument responses.

This is now mentioned in line 120-127: "The dilution ratio was set at 1:17 during the car emission experiments and 1:60 during the RWC experiments because of the large difference in PM and gas phase concentrations of the two PM sources and to assure a final maximal PM concentration between 500 and 2000 □g/m$^3$ (Fig. 2 and 5)."

Page 5, Line 141-142: please provide the reasons for different start times for OPROSI and OOPAAI.

The reasons were added (line 155-157): "The OOPAAI was started the day before an experiment day and blank measurements were run overnight to assure a stable blank. The OPROSI was started only one hour before the start of an experiment as the higher flow rates lead to a stable blank within a shorter time."

Page 5, 142-143: please provide a brief explanation of what the "filters" are used for and include a reference where more details can be found. Also, filters were changed for RWC experiments and only with OPROSI. While the reasons are provided in the results section, it is useful to

make a reference here for clarity (e.g. 'see section x for details').
We changed the sentence to clarify this (line 157-158): "The cellulose grade 1 filters inside the liquid systems of both instruments, which remove insoluble particles were changed daily to avoid excessive contamination by insoluble particles (see section 3.2 for further details)."

Page 5, Line 144-145: please provide a brief explanation of how the calibration was performed, and the rationale for calibrating the instruments once a week and not more often. For offline analysis, calibration is performed per sample batch, which happens daily, if not more often.
We changed the sentences to clarify that (line 159-161): "The OOPAAI and OPROSI were calibrated once a week to ensure consistent performance for the duration of the measurement campaign, as described in Wragg et al., 2016 (Wragg et al., 2016) and Utinger et al., 2023.(Utinger et al., 2023)» The instruments run very stably over several days which we tested extensively in Wragg et al., 2016, and Utinger et al., 2023, and therefore more frequent calibrations were not necessary.

Page 6, Line 153: what is the rationale for choosing pH 6.8? The typical pH in the lower respiratory tract under normal conditions is just above 7.
You are right that in a healthy lung the pH is around 7.4. The very slightly acidic conditions used in our instrument were chosen due to a tradeoff of an increase the stability of DHA, which is described in detail in Utinger et al. 2023 (https://doi.org/10.5194/amt-16-2641-2023)

Page 6, Line 172-174: There was no mention of ELPI (and other aerosol instruments used in this study) until this point (see my earlier comments). Also, which data was "averaged to 10s"? the AA and ROS data (they have 10-min resolution)!
See the answer and changes to the text in the general section about the use of an ELPI. The time resolution of OOPAAI and OPROSI is about 10min, but both instrument record the detector signal at a higher frequency of 1sec so that also information of the transient conditions can be acquired. To match the OOPAAI and OPROSI data to the ELPI, their data was averaged to 10s.

Page 7, Line 179: regarding particle densities, Paul et al. mentions that 1.6 g/cm3 was "assumed". What was the rationale for choosing this value in this study? This is important, as the value is a basis for calculating EFs.
The publication Paul et al., 2024, analysed experiments from the same campaign and we therefore use the same particle density as in this peer-reviewed paper.

Page 7, Line 182-184: regardless of the reference provided, a description of the method used for calculating EFs should be provided in the supplement and referenced in the text (specially considering that this aspect forms a section in this manuscript).
We changed the wording and added the equation used method to the following (line 198-200): " The methods to derive ROS and OP emission factors (EF) for car exhaust and for RWC PM are described in Paul et al., 2024 and Reda et al., 2015, respectively. (Paul et al., 2024; Reda et al., 2015) and calculated as shown in the following equation:

$$EF = \frac{nmol\ ROS}{kg\ fuel} = \frac{mg\ aerosol}{kg\ fuel} \ x \ \frac{nmol\ ROS}{mg\ aerosol} \text{,,}$$

Page 7, Line 195: please include the detection limits of the online instruments. We added the following sentence to the manuscript to clarify that (line 218-220): "LODs (in units of nmol DHA m-3 and nmol H2O2 m-3, respectively) depend on the OP and ROS content of the respective particle and is therefore variable. For SOA, LOD is around 5ug/m3 (Utinger, Wragg et al.) which is much lower than any of the conditions measured during this campaign where no GFP was used .

Page 7, Line 202-204: a note should be added describing which load conditions generated the highest and lowest OP and ROS as well as PM mass (despite being evident in the figure). Added a sentence to give this information (line 229-230): "The highest mass concentrations and also OP and ROS signals were measured at 50 km/h compared to 80 km/h and 100 km/h were lower OP, ROS and mass concentrations were observed."

Page 8, Figure 2: The figure's key should follow the color bars in the figure, starting with idling, followed by 50, 100, and 80 km/h. Also, it would be useful to include the data for when PEAR was offline.

We changed the graph so that the figure's key follow the color bars. Because we measure only OP and ROS of particulate matter and because there was a particle filter in the car, no primary PM emissions occurred and therefore no OP and ROS of primary car emissions was measured. However, the raw data is shown in the SI as stated in line 56.

Page 8, Line 214-217: statistical significance should be included when comparing the results. The fast changing driving cycles and the variability of the individual repeat cycles made it not possible to measure stable plateaus where we could have compared single conditions and derive significant differences between the various engine loads. As already discussed further up we therefore modified the manuscript at several places so that the claim is less strong and we only indicate a trend.

Page 9, Line 243-245: comparison of short vs. medium ageing should be performed with statistical significance (considering the variabilities with the observations). Also, authors should include an explanation for the higher variability seen between short and medium ageing during each load cycle.

A statistical significance could not be determined as explained above.

Unfortunately, we could not rationalise the difference in OP and ROS between the two aging conditions.

Page 9, Line 249-251: the sentences need rephrasing. It is not entirely clear what the authors are implying here.

This sentence was deleted.

Page 10, Line 284-286: discussing changes should include values, e.g. increased or decreased from x to y, or x-fold, etc. Qualitative discussion is not appropriate. Also, please mention the actual batch-to-batch variability seen with RWC experiments (in numerical terms).

Changed the sentence to clarify (line 302-306): "$OP_V$ and $ROS_V$ reached peak values in the first half of a 35 min long batch with approximately a factor of ten higher compared to values at the end of a batch. This again demonstrates how both online instruments are capable of capturing fast changing emission characteristics. After six batches of wood added to the stove, the ember phase started, characterised by low emissions of $OP_V$ and $ROS_V$ as well as particles."

The standard deviations in Figure 6 and 7 already show the batch-to-batch variability throughout a batch.

Page 11, Line 287-288: "In contrast to the car emissions, both instruments responded to the primary RWC emissions due to the significantly higher primary PM emissions": Differences here should be stated in numerical terms (e.g. significantly higher?).

Changed the sentence for clarification (line 309-310): "In contrast to the car emissions where no primary particles were emitted due to the GPF, both instruments responded to the primary RWC emissions."

Page 11, Line 290: Figure S6 does not exist in the supplement. Is this referring to Figure S5??
Corrected wrong figure labeling in the SI

Page 11, Line 297: "Figure S 2 shows a clear decrease of the ROS signal…" The figures in the supplement should be rearranged to follow the order of appearance in the main text. It is not the case in this manuscript. This needs to be addressed as well as the following issues:
Changed

- Provide the full caption for Figure S2
Fixed layout mistake.

- There are two Figures labeled S1 in the supplement (second one appearing after Figure S3)!
Corrected wrong figure labeling in the SI

Page 11, Line 297-299: "…clear decrease of the ROS signal with higher concentrations of charcoal." This is a clear limitation for OPROSI and should be mentioned in the manuscript.

To clarify that a sentence had be added (line 320-322): "Figure S 6 shows a clear decrease of the ROS signal with higher concentrations of charcoal, which is a fundamental limitation of the DCFH assay at very high insoluble particle concentrations."

Page 12, Line 302: "Photochemical ageing led to a significant increase…" How significant? Statistically significant or else? This should be expressed in numerical terms and include a statistical assessment.

Line 333 This introductionary sentence was deleted in the revised version as it did not add to clarity.

Page 12, Line 307: "…resulted in low and stable OPM values in the same range as the car emissions…" Please mention the actual values/ranges.

The following was added (line 329-330): "The primary aerosol (**Error! Reference source not found.**A) resulted in low (between 0 and 2 DHA nmol $\mu g^{-1}$) and stable $OP_M$ values in the same range as the car emissions for the majority of a batch."

Page 13, Line 325: "…it was observed that more ageing does not induce a higher OP." This is not entirely true. It depends on various factors including the assay type. Indeed, some studies found an increase in OP (with thiol antioxidants) with OH oxidation or organic matter up to 8 days. This aspect should be commented on.

We now clarify for which assay this statement is correct (line 347-349): "Also for certain SOA types from gaseous precursors β-pinene and naphthalene it was observed that using the DCFH assay more ageing does not induce a higher OP. (Offer et al., 2022)"

Page 13, Figure 7: If longer aging of RWC emissions results in reduced OP (Figure 6) but increased ROS (Figure 7), should one conclude that peroxides have negligible effect on AA oxidation? The authors should provide a short discussion on this.

This is a possible interpretation of the results but, unfortunately, without having detailed chemical analysis, which was not the focus of this study, it is difficult to make such statements. Therefore, we decided to rather not mention such speculative conclusions in this paper.

Page 14, Line 383-385: "EFOP of both aging conditions (blue and blue stripes for the car and short and medium for RWC) are very similar for both sources." The sentence needs a better phrasing (mentioning the actual ageing conditions, providing numbers, and comparing values with statistical significance).

Sentence was changed to (line 409-410): "No clear trend in $EF_{ROS}$ between the two aging conditions (blue, short aging and blue stripes, medium aging for the car and short and medium for RWC) was detected."

Page 15, Line 384-385: "For EFROS and EFOP gasoline car emissions are up to 8 times higher than RWC values, which is the opposite compared to mass normalized OPM and ROSM concentrations." What is the reason for this counter intuitive observation. This reiterates the need to describe in the text how EFs were calculated; this would make it easier for the reader to understand the observations.

We added the calculations of the EF to the manuscript (see comment above) to try to clarify that.

The metric of $EF_{OP}$ and $EF_{ROS}$ is to our knowledge published for the first time in this manuscript, so it is difficult to rationalise and compare with other values.

We changed the end of that paragraph to clarify that to the following (line 410-412): "$EF_{ROS}$ and $EF_{OP}$ for gasoline car emissions are up to 8 times higher than RWC values, which is the opposite compared to mass normalised $OP_M$ and $ROS_M$ concentrations. This can be explained in part by the very different units used for normalising $OP_M$ and EF, respectively. For $OP_M$, the mass of emitted particles is used to normalise OP values, whereas for EF the mass of burnt fuel is used."

Page 16, Line 416-417: I do not follow the logic behind the closing sentence. The study certainly sheds light on dynamic processes behind OP observations but how does it provide weight of evidence that OP should be considered as a valid metric to evaluate health effect of PM. I do not think this study touches that area. Please revise and provide an appropriate closing statement.

We changed the closing statement to the following sentence (line 443-445):

"Overall, the results presented here show the importance of measuring OP and ROS with a high-time resolution to capture their dynamic nature and to consider different atmospheric ageing times of these potential PM toxicity markers."

**Minor/Technical comments:**

Page 2, Line 60: change "while simultaneously deplete" to 'and to deplete' or 'simultaneously depleting'
Changed

Page 2, Line 61: change "linked to causing negative health effects" to 'linked to negative health effects'
Changed

Page 3, Line 98: change "potentially allowing to assess health risks" to 'enabling the assessment of health risks'

Changed

Page 4, Line 105: add the abbreviation for "o-phenylenediamine"

Changed the whole paragraph to make it more consistent (line 109-114): All chemicals were obtained from Sigma-Aldrich and were used without further purification unless otherwise indicated: ascorbic acid (AA, 99.0%), dehydroascorbic acid (DHA, 99.0%), hydrochloric acid (HCl, 0.1 M) sodium hydroxide (NaOH, 0.1 M) solution, Chelex 100 sodium form, o-phenylenediamine (OPDA, ≥99.5%), 4-(2-hydroxyethyl)-1-piperazineethanesulfonic acid (HEPES, ≥ 99%), methanol (99.9%), peroxidase from horseradish (Type VI, HRP), 2,7-Dichlorofluorescin diacetate (DCFH-DA, 97%), 3% hydrogen peroxide solution, phosphate-buffered saline solution (PBS, 1M), zero grade air (Model 737–250, Aadco Instruments Inc., USA), $N_2$ gas (99.999%, Linde, Finland).

Page 5, Line 120-121: change "…to measure both primary and secondary emissions." to '…to measure OP and ROS from both primary and secondary emissions.'
Changed

Page 6, Line 157: For clarity, please indicate that the text is now discussing the ROS measurement. The same should be done for the AA assay.

Line 165 for the AA assay was changed to: "For the ascorbic acid assay the water was passed through a fritted column filled with 100 g Chelex 100 resin, to further reduce the amount of contamination."
Line 174 for the DCFH assay was changed to: "For the ROS measurements of the DCFH assay the HRP stock solution was prepared weekly by dissolving 5000 units in 500 mL of water and stored at 4 °C. HRP working solution was prepared using 100 mL of HRP stock solution, 100 mL of 1 M PBS, and 800 mL of pure water."

Page 7, Line 196: change "…OPV and ROSV concentrations…" to "…OPV and ROSV values…".
Changed

Page 11, Line 289: I suppose the authors are referring to Figure 6a when saying "…during the aged experiments (6)…". Please revise.

Changed to: "The OOPAAI observed an $OP_v$ signal for primary emission which was up to three times lower than during the aged experiments (Figure S6), which can be expected as ROS are predominantly formed through oxidation processes."

Page 12, Line 310: "The average OPM values…" Please mention the ageing condition this refers to.

Added: "For the short ageing,"

Page 12, Line 312: change "The large increase in OPM…" to 'For medium ageing condition, the large increase in OPm…'

Changed

Page 12, Line 314: "…similar to short aging conditions (Figure 6C)." I think this is referring to Figure 6B. Please revise.

Corrected

Page 12, Figure 6: for clarity, labels should be added to the individual plots to indicate various ageing conditions.

Added as suggested.

Page 13, Figure 7: The plots are labeled as A and B while the caption refers to B and C. Please revise.
Corrected

Figure S1 in the supplement (Mass concentrations measured by the SMPS and ELPI during a stove (A) and car (B) exhaust measurement): It is nearly impossible to read the color coding for PM0.3, 0.9, 2.2, and SMPS. This needs improvement. Also, please number this figure correctly (in the order of appearance in the text).

Figure has been improved and labeling mistake has been corrected.

**Answers Reviewer 2**

In this paper titled "Emission dynamics of reactive oxygen species and oxidative potential in particles from a gasoline car and wood stove", authors report studies related to the OP and ROS activity of particles of gasoline passenger car emission and of residential wood combustion (RWC), both generated as fresh and photochemically aged (to simulate the atmospheric ageing), by using two different online instruments. The reported tests to measure OP and ROS being AA and DCFH assays established. Based on these analysis Authors stated that no primary particle was observed in fresh gasoline car emission, probably thanks to the presence of a gasoline particles filter (GPF) to reduce primary exhaust emissions in 6d gasoline car. Otherwise an increment in OP and ROS particles were observed after photochemically aging. For RWC the increase of OP and ROS contribution is also observed between fresh and aged RWC. The increase of OP and ROS was associated in both samples to the formation of secondary particles due to the photochemical ageing. In my opinion the experiments are of interest to give information about the chemical effects of the photochemical aging and the experimental set-up is well designed.

The authors like to thank the reviewer for their constructive comments. Our answers are given below in blue font.

However, I will point out that there are some drawbacks to be improved before accepting for publication. In particular, the experimental set-up has been reported and discuss from the authors almost in two previous papers. In the present paper the authors just apply the same set-up to obtain two different anthropogenic sources of particles generated to studies their OP and ROS properties at different aging level.

We understand the concern of the reviewer, but we believe that for a novel technique, as our online ROS and OP analysis, it is crucial to demonstrate its versatility and applicability to a variety of scenarios. In particular, by employing the same experimental setup, we can ensure consistency and comparability between studies while expanding our understanding of the technique's potential. Furthermore, this approach allows us to build on the findings of our previous work, providing a more comprehensive perspective on the capabilities of the instrument and of OP and ROS as a novel air pollution metric in general and as a metric for emission assessments more specific.

In this study, the focus is placed on measuring two distinct anthropogenic sources of particles, specifically chosen to explore their oxidative potential (OP) and reactive oxygen species (ROS) characteristics at different aging stages. While the setup remains consistent with our earlier studies, the novelty lies in the characterization of these specific particle sources and the insights gained with different aging conditions. We measured, for the first time to our knowledge, highly dynamic anthropogenic sources using online OP and ROS instruments.

To strengthen the results, they need to correlate the OP and ROS outcomes to the chemical composition of both particles typologies by experimentally measuring the chemical components of particles also at different level of aging (i.e. carbon and metal contents). In my opinion there is a lack of chemical information's to be filled to better define and discusses these indicators.

We agree that it would be important to have more chemical information, but unfortunately this is not available for this campaign. The focus of this paper is on the methodology with the two different ROS and OP instruments being applied in the same campaign to important urban PM emission sources, the comparison of the two different aerosol types and aging

conditions and illustrating the highly variable and highly dynamic nature of PM and ROS from these sources, which can only be assessed with online instruments as the ones used in this study.

Finally, for completeness I suggest to cite in the introduction an appropriate recent published paper titled: "Characterisation of the correlations between oxidative potential and in vitro biological effects of PM10 at three sites in the central Mediterranean" from Guascito et al., 2024 (Journal of Hazardous Materials Volume 448, 15 April 2023, 130872), were was investigate the correlations among acellular and intracellular toxicity indicators, the variability among the sites, and how these correlations were influenced by the main sources by using PMF receptor model coupled with MLR.
We added the citation in line 65.